# AttG-BDGNets: Attention-Guided Bidirectional Dynamic Graph IndRNN for Non-Intrusive Load Monitoring

**Zuoxin Wang and Xiaohu Zhao ***

School of Information and Control Engineering, China University of Mining and Technology, Xuzhou 221116, China
* Correspondence: zhaoxiaohu@cumt.edu.cn; Tel.: +86-133-0520-6662

**Abstract:** Most current non-intrusive load monitoring methods focus on traditional load characteristic analysis and algorithm optimization, lack knowledge of users' electricity consumption behavior habits, and have poor accuracy. We propose a novel attention-guided bidirectional dynamic graph IndRNN approach. The method first extends sequence or multidimensional data to a topological graph structure. It effectively utilizes the global context by following an adaptive graph topology derived from each set of data content. Then, the bidirectional Graph IndRNN network (Graph IndRNN) encodes the aggregated signals into different graph nodes, which use node information transfer and aggregation based on the entropy measure, power attribute characteristics, and the time-related structural characteristics of the corresponding device signals. The function dynamically incorporates local and global contextual interactions from positive and negative directions to learn the neighboring node information for non-intrusive load decomposition. In addition, using the sequential attention mechanism as a guide while eliminating redundant information facilitates flexible reasoning and establishes good vertex relationships. Finally, we conducted experimental evaluations on multiple open source data, proving that the method has good robustness and accuracy.

**Keywords:** non-intrusive load decomposition; global context semantics; attention guidance mechanism; dynamic aggregation; adaptive graph topology

## 1. Introduction

Electricity has become an indispensable resource for human life with the increase in social population and the continuous progress of various industries. However, due to the reduction and depletion of fossil resources, means of effectively saving electricity resources and preventing waste have become of great concerned. In the power saving process, power consumption management and energy efficiency optimization are essential: non-intrusive load monitoring (NILM) [1] is used to estimate the power demand of each device in the case of monitoring the total power demand signal recorded by a single meter of multiple devices, to formulate corresponding energy-saving measures, that is, it is adequate to improve energy consumption management and energy efficiency analysis to achieve maximum cost savings. In addition, non-intrusive load detection can more accurately understand the composition of users' electricity consumption, help power companies formulate electricity prices and allocate resources scientifically and rationally, and at the same time, provide a more accurate basis for planning and scheduling issues in the power system. In addition, non-intrusive load monitoring (NILM) can prompt electrical appliance users to intuitively understand the power consumption and operation of various equipment in different periods and regulate and adjust power consumption behavior.

In the early stage, to achieve accurate non-intrusive load monitoring (NILM), most of them used simple machine learning methods for optimization analysis, such as k-nearest neighbor (k-NN) [2], support vector machine (SVM) [3], matrix decomposition [4], etc. These methods mainly implement power measurement by sampling within intervals of

seconds or minutes. Hidden Markov models [5] benefited from the conversion of consumption levels that simulated the actual energy consumption of the target equipment and were widely used in the early stages. Although the early methods were energy-saving, they required manual participation in feature setting and screening. Their accuracy often depended on human subjectivity, so they heavily relied on previous expert experience and were challenging to apply on a large scale. In recent years, the successful application of deep learning technology in many fields also provides a new idea for non-intrusive load monitoring (NILM). For example, the convolutional neural network [6] automatically learns practical details by changing the activation state of neurons, overcoming the need for human participation in feature selection, and the setting is widely used. Nonetheless, convolutional networks perform poorly in the establishment of long-term dependencies. Long short-term memory [7] and recurrent neural networks [8] have attracted attention because of their ability to learn long-term relationships; that is, they can learn long-term distance information on sequence data. At the same time, to avoid redundant information being reused, forgetting gates are used to achieve memory selection. Although these methods maximize information and reduce errors caused by manual participation, capturing details from signals in non-intrusive load monitoring (NILM) is difficult. There is a lack of sufficient knowledge on users' electricity consumption habits which results in the poor accuracy of monitoring or decomposition models and suboptimal generalization performance. To address these issues, we propose an attention-oriented bidirectional dynamic graph IndRNN method, which aggregates and transfers node information by building a topological graph, and extracts device-specific power usage from aggregated signals, aiming to perform accurate non-intrusive load monitoring (NILM).

The main contributions of this study are listed below.

- A new attention-guided bidirectional dynamic graph IndRNN method (AttG-BDGNets) is proposed, which is the first attempt to enhance node representation in the form of dynamic aggregation in the NILM task, and utilize node aggregation and transfer capabilities to explore the relationship between equipment and power.
- Model the NILM sequence through a bidirectional independent recurrent neural network and establish long-distance dependencies while learning contextual semantics. Furthermore, utilize the local attention guidance layer to enhance the feature representation through the complementary relationship between a dynamic graph and temporal features.
- The designed weighted loss function optimizes the dynamic graph and the bidirectional independent recurrent neural network separately so that each branch can obtain the optimal representation. It is worth noting that the vertex relationship (the edge between nodes) is jointly calculated by the planar Euclidean distance and the spatial cosine similarity; that is, the vertex relationship is explored from both temporal and spatial aspects. Finally, evaluation and verification were performed on two baseline datasets, REDD and UK-Dale, and the best prediction and classification performances were achieved.

The remainder of this article is organized as follows: Section 2 mainly reviews work related to non-intrusive load monitoring (NILM). Section 3 focuses on the proposed AttG-BDGNets monitoring framework and gives the working principles and functions of different components; Section 4 provides experimental results and analysis and discusses and visualizes the experimental results; the summary and future research plan are given in Section 5.

## 2. Relate Works

In the early stage, the non-intrusive load monitoring (NILM) methods mainly focused on simple machine learning. For example, Gong F. et al. [3] proposed a PSO-based-SVM's non-intrusive load monitoring load decomposition method and used the power difference method to detect the switch state of electrical equipment, and at the same time, used the multi-feature classification (MFC) function of PSO-SVM to identify the switch state of elec-

trical equipment. To obtain the energy consumption of electrical appliances in buildings, Liu H. et al. [9] proposed a weighted current harmonic vector to increase the weight of valuable harmonics and geometrically calculated the harmonic vector so that all information in the feature can be preserved; and established a decomposition model based on multi-objective particle swarm optimization is proposed, and the error sum and standard deviation are used as objective functions. To realize non-intrusive load monitoring and decomposition from two aspects of load identification and load decomposition, Lin J. et al. [10] first based their work on the load characteristics of the database and used the load decision tree algorithm to analyze and identify the equipment composition of the mixed electrical equipment group. Li Z. et al. [11] proposed a new clustering decomposition algorithm to solve the problems of low non-intrusive load decomposition capability and low precision when two electrical appliances start and stop simultaneously. Firstly, the measured power was analyzed, and DBSCAN was used to analyze the filtered noise. At the same time, the remaining power points are clustered through the adaptive Gaussian mixture model to obtain the cluster centers of electrical appliances. Finally, the corresponding current waveforms are correlated to improve the recognition accuracy. Lin Y. H. et al. [12] proposed a particle swarm optimization (PSO)-based DSM NILM system to effectively monitor and manage industrial, commercial, and residential electricity loads. To realize the load decomposition of the non-intrusive load monitoring system, Xiao Y. et al. [13] proposed a load event matching method based on graph theory based on the improved Kuhn–Munkras algorithm. Focusing on the low accuracy of non-intrusive load monitoring in traditional statistical methods, Mao Y. et al. [14] proposed a non-intrusive load monitoring model based on conditional random field (CRF). Based on user power consumption data, the linear conditional random field generates the state characteristic function and state transfer function. Although these simple machine learning methods can improve the accuracy and efficiency of non-intrusive load monitoring, they require manual participation in feature design and screening and heavily rely on the experience and knowledge of experts. It is challenging to meet the growing application requirements, and at the same time, it is time consuming and expensive, with low monitoring accuracy and poor applicability.

In recent years, deep learning methods have received extensive attention in many fields, such as medical image generation and detection, relying on their strong self-learning ability [15–18]. Many researchers have been inspired to apply these techniques to NILM tasks. For example, Kelly et al. [19] applied deep neural networks to non-intrusive load monitoring tasks and achieved accurate predictions, then used memory networks such as LSTM to process the high-frequency time series to predict the start time, end time, and average power demand of each device. Considering the problems of explosive gradients, vanishing gradients, and network degradation in deep neural networks, Jia Z. et al. [6] proposed a non-intrusive load monitoring method based on bidirectional extended convolution for learning low-frequency data, mainly relying on temporal convolution. The residual block of the network solves the problem of vanishing and exploding gradients. At the same time, expanding the receptive field encourages the network to learn better feature representations to improve the model's overall performance. Hwang H. et al. [7] found that, when non-intrusive load monitoring is performed on low-frequency data, the power usage patterns that change over time will disappear. Features cannot be correctly acquired to classify devices, and a new non-intrusive method is proposed. A load monitoring model that can learn datasets with unbalanced data classes utilizes long short-term memory (LSTM) to extract features and improve the feature representation ability of LSTM through predicted feedback. Zhang et al. [20] proposed an energy-decomposed sequence-to-point learning method, where the midpoint of the device window is regarded as the classification output of the neural network, and the aggregation window is used as input. Shin et al. [21] proposed a sub-task gating network (SGN) that combines two CNNs, a regression sub-network, and a classification sub-network for a non-intrusive load monitoring task. Wei J. et al. [22] proposed a method for applying work state mining and sequence transition models to non-intrusive load monitoring. First, we determined the operating power of each appliance

in different operating states and generated a combined code to represent the operating states of all appliances. Then, the total energy consumption signal and combined status codes are trained on a sequence-to-sequence model that considers run-time dependencies. The model integrates the time scale information and signal amplitude characteristics of the power working state and converts the power consumption into a state code for load decomposition. Jaramillo et al. [23] aimed to fill the gaps in load modeling and NILM knowledge, providing deri with information for the "green deal" transition, and supporting standardization. Kong W. et al. [24] proposed a practical and effective non-intrusive load monitoring method for estimating the energy consumption of commonly used multifunctional appliances (Type II appliances). Considering the fact that the traditional non-intrusive load monitoring algorithm has long-term problems such as a high misjudgment rate and low precision of decomposed power values, a non-intrusive load with an attention mechanism based on a sequence-to-sequence (seq2seq) model is proposed, namely a monitoring model. The model first embeds the input active power time series into a high-dimensional vector, extracts information with a long short-term memory (LSTM)-based encoder, selects the most relevant information for decoding, and finally, disaggregates the decoder wrapped with an attention mechanism result. Rafiq H. et al. [8] proposed an NILM algorithm, which uses data augmentation to generate synthetic data for training the deep convolution of each target device due to the low accuracy of decomposition of new unseen data, which is unsuitable for practical applications. Neural network models also devise an evaluation method that relies on the device-predicted total and ground-truth energies to provide details on the algorithmically predicted total overlapping energies, missing energies, and extra energies. Moradzadeh A. et al. [25] combined the Laplacian feature map (LE), convolutional neural network (CNN), and recurrent neural network (RNN) to transfer the significant features and specific values of the energy consumption curve of household appliances to a low-dimensional space and use the recurrent convolutional network to improve the structure of the fully connected layer significantly CNN, so that there is no over-fitting problem in the identification and separation of HEA types, and it has high accuracy. Nie Z. et al. [26] used a sequential point deep neural network and constructed a comprehensive strategy non-intrusive load monitoring technology based entirely on the deep feature-guided attention mechanism. Faustine A. et al. [27] considered that the performance of the device classifier is highly dependent on the signal characteristics of the load, but it is difficult to effectively distinguish between similar signals, which increases the difficulty of subsequent device classification, and proposed a weighted cycle graph (WRG) for representing these signals and improve the performance of device classification. Jiao X. et al. [28] considered that many existing research methods have specific load decomposition errors when dealing with multi-modal devices. A non-intrusive load monitoring model based on a graph neural network is proposed. The graph structure is used to represent the relevant information between data nodes, combined with the long-term and short-term memory to extract the data's time-domain characteristics while retaining the power data's time characteristics. The correlation between different modes of equipment is improved, and the decomposition error is reduced. Jie Y. et al. [29] proposed a non-intrusive load decomposition model based on a graph convolutional network (GCN) by fully mining the user's electricity consumption habits. The model first uses the time features extracted from the user's electricity consumption habits and constructs the power sequence as graph data based on the spectral graph theory as network input; then, based on the graph convolutional neural network, extracts the power attribute characteristics of each electrical appliance and its correlation time-dependent structural features for non-intrusive load decomposition. These methods effectively improve the representation of key information in non-intrusive load monitoring tasks, but there is a lack of interaction between different information. At the same time, they aim to explore high-level semantic features while ignoring the rich semantics contained in low-level features so that when obtaining the high-order features of the model, which cannot be effectively represented, the final recognition and decomposition accuracy will be affected.

## 3. Proposed AttG-BDGNets Approach

In this section, we focus on AttG-BDGNets to improve the performance of non-intrusive load monitoring (NILM). First, the problem of non-intrusive load monitoring is formulated, and the basic principles and architecture of the network are introduced. Following this, the individual modules of the proposed method will be described in detail.

### 3.1. Non-Intrusive Load Monitoring Problem

Non-intrusive load monitoring (NILM) was first proposed by George W. in 1992. Assuming that the total power consumption of all devices is $(x_1, \cdots, x_T)$, there are $M$ metered devices under test and $K$ unmetered devices, $T$ is the sampling time indicating the length of the input sequence; the main task of our proposed AttG-BDGNets is to decompose them from $M$ and $K$ device contribution. It is worth noting that the total power consumption consists of the power consumption of all devices (both metered and unmetered) and some unknown noise terms. The total power consumption at time t for a set of metered $M$ and unmetered devices $K$ is shown in the equation.

$$x_t = \sum_{i=1}^{M} y_t^i + \sum_{j=1}^{K} (z_t^j + \varepsilon_t) \tag{1}$$

where $y_t^i$ represents the power consumption contribution of the $i^{th}$ device under test at time $t$; $z_t^j$ represents the power consumption of the $j^{th}$ unmetered device at time $t$; $\varepsilon_t$ represents the power consumption contribution of the noise term at time $t$.

### 3.2. Overview

The overall network structure of AttG-BDGNets is shown in Figure 1. The proposed AttG-BDGNets monitoring method consists of two essential modules: the dynamic graph convolution module (DynamicGCM) and the attention guidance module (AGM). The dynamic graph convolution module (DynamicGCM) aims to gather and transmit functions to obtain the hidden dynamic information in the NILM sequence. Simply put, the load actions included in the NILM sequence have good steady-state and transient performance; that is, their transition process should be taken into account when obtaining transient characteristics and the instantaneous power and switching transient characteristics, so the extraction process of steady-state features is more accessible to obtain than transient features. The dynamic graph convolution module (DynamicGCM) can capture this dynamic information and use the dynamics of the inter-layer neighborhood. The aggregation strategy learns node representations from the graph and directly associates distant correlation graph nodes to effectively learn longer temporal dependencies, forming an effective interaction between steady-state features and transient features. AGM mainly includes a local attention guidance layer and a bidirectional independent recurrent neural network module, in which the bidirectional independent recurrent neural network (Bi-IndRNN) models the power consumption data sequence in different periods from the positive and negative directions and obtains contextual semantic details, fully capturing the correlation between changes in electricity usage data and household members' electricity usage behavior. The local attention guidance layer (LAG) combines the global features captured by the dynamic graph convolution module (DynamicGCM) with the contextual semantics acquired by the bi-independent recurrent neural network (Bi-IndRNN), forming a complementary relationship between them to improve the semantic representation ability, which is when the representation effect of $f_\varsigma$ is not good. This guide layer is used to highlight the representation of $f_{ir}$ and prevent $f_\varsigma$ from affecting $f_{ir}$. On the whole, we first process the input NILM sequence and input the initial dynamic graph convolution module (DynamicGCM) and bidirectional independent recurrent neural network (Bi-IndRNN), respectively, to obtain the corresponding feature semantics $f_\varsigma$ and $f_{ir}$, and use the local attention-guided layer to perform the feature fusion to strengthen semantic representation; finally, accurate prediction is achieved.

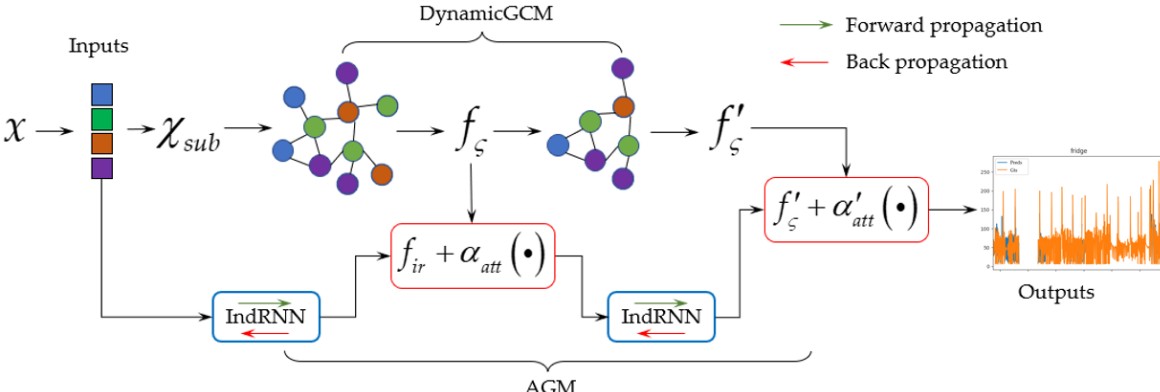

**Figure 1.** The overall network structure of AttG-BDGNets. The IndRNN in the blue box represents a bidirectional independent recursion neural network (Bi-IndRNN), the green rightward arrow represents the forward propagation of IndRNN, and the red leftward arrow represents the backpropagation of IndRNN; $f_{ir}$ represents the output characteristics of the initial layer bidirectional independent recursion neural network; $\alpha_{att}(\cdot)$ and $\alpha'_{att}(\cdot)$ represent the local attention embedding module we designed and the local attention guidance layer, respectively; $AGM$ represents the attention guidance module; $f_\varsigma$ and $f'_\varsigma$ represent the first and second layer outputs of the dynamic graph convolution, respectively; $DynamicGCM$ represents the dynamic graph convolution module; $x$ represents the input sample; $\chi_{sub}$ represents the reconstruction of $x$, which is the input of the dynamic graph convolution module; "*Outputs*" and "*Inputs*" represent the resulting output and input, respectively.

*3.3. DynamicGCM*

Graph convolution (GCN) [30] is a network that can directly perform convolution operations on topological graphs and mainly achieves feature capture by converging node features [26]. The dynamic graph convolution module we designed not only learns node representation through node aggregation and transfer functions but also realizes dynamic aggregation between layer neighborhoods, which promotes the correlation between all devices and total power consumption, as well as a correlation between the unmetered device and the metered device under test. In addition, a dependency relationship may also be formed between associated nodes that are far away.

Assume that the metered device under test is $M$, the unmetered device is $K$, the total power sequence of all devices is $x = \{x_1, \cdots, x_\chi\}$, $T$ is the sampling time, the sliding window length is $L$, and the *ith* power sequence is $x_i = \{x_i, \cdots, x_L\}$. Secondly, the sliding window is moved by *jth* sampling time points. Finally, $T - L + j$ power sequences are obtained so that each power sequence can be represented as a topological graph, where there are $N$ nodes in the graph, that is, $N$ devices contribute to each power sequence. There is a correlation between the nodes, and this correlation is called the edge weight between nodes. To put it simply, we build a topology map with power sequences. Each power sequence has several electrical devices that generate power in different periods, and each device is a node of the topology graph. Formally, the topological graph $\varsigma$ is shown in Equation (2).

$$\begin{cases} \varsigma = (\nu, \epsilon) \\ \nu = \{\nu_1, \cdots, \nu_N\}, N \le M + K \\ \epsilon = \{\epsilon_1, \cdots, \epsilon_N\} \end{cases} \tag{2}$$

where $\nu$ represents the set of nodes, and $\epsilon$ represents the set of edges.

In order to obtain more efficient associations and achieve accurate predictions, at sampling time point $t$, we performed an optimization operation on the topology map, as mathematically shown in equation.

$$\begin{cases} f_\varsigma = \sigma(\tilde{D}^{-\frac{1}{2}}\tilde{A}\tilde{D}^{-\frac{1}{2}}\chi_{sub}\Theta) \\ \tilde{A} = A + I \\ \tilde{D} = \sum \tilde{A} \end{cases} \tag{3}$$

where $\tilde{D}$ represents the degree matrix of the adjacency matrix $A$; $I$ represents the identity matrix; $A$ represents the adjacency matrix, which is composed of nodes and edges; $f_\varsigma$ represents the output features of the first layer of graph convolution; $\tilde{A}$ represents the renormalization of the adjacency matrix $A$ Lass processing; $\Theta$ represents the weight matrix; and $\chi_{sub}$ represents the input power sequence.

Assuming that there are devices $m$ and $n$ at the sampling time point $t$ and power sequence $\chi_{sub}$, that is, nodes $v_m$ and $v_n$, if there is an association between them, an edge will be formed, and if there is no correlation, there will be no edge. At the same time, to establish an effective dependency and association relationship between them, it is necessary to promote the network to learn a better node representation. We take the weighted mean of the planar Euclidean distance and spatial cosine to compute the edge weights. The calculation process is shown in the equation.

$$A_{m,n} = \begin{cases} \frac{1}{2}[||v_m - v_n||_2^2 + \frac{v_m \cdot v_n}{||v_m||||v_n||}] & v_m \neq v_n \\ 1 & v_m = v_n \\ 0 & otherwise \end{cases} \tag{4}$$

where $|| \cdot ||_2$ indicates two paradigms, which is equivalent to the plane Euclidean distance; $|| \cdot ||$ indicates the modulus of the characteristics of node $v_m$; and $v_m = v_n$ indicates that the node is a self-loop node.

Although the traditional node aggregation method effectively aggregates the structural information of adjacent nodes, it causes the captured semantic information to spread only in a local area. It is difficult to fully capture the dependencies between distant nodes. That is to say, in the classic graph convolutional network (GCN), the graph is fixed throughout the convolution process, which reduces the representation performance of the nodes. Therefore, we realize the aggregation of inter-layer information through dynamic aggregation. The topological graph structure can be seen here. The process is gradually refined, and the node representation is further optimized. In addition, this aggregation method can fuse the current embedded features and the graph information used in the previous layer, which is used to improve the ability of the graph structure to capture global details. The dynamic aggregation process is shown in equation.

$$f'_\varsigma = \sigma(\tilde{D}^{-\frac{1}{2}}\tilde{A}\tilde{D}^{-\frac{1}{2}}f_\varsigma\Theta') + \lambda f_\varsigma \tag{5}$$

where $\lambda$ represents the parameter factor, which is set to 0.35. $\sigma(\cdot)$ indicates the activate function of LeakyReLu.

### 3.4. AGM

AGM mainly consists of a bidirectional independent recurrent neural network (Bi-IndRNN) [31] and a local attention [32] guidance layer. Among them, Bi-IndRNN models the input NILM sequence from both positive and negative directions, aiming to explore the correlation between the total power and all devices and to construct the long-distance dependence between them in different periods. The traditional recurrent neural network (RNN) cannot effectively solve the problems of gradient disappearance and gradient explosion during the training process, and it is not easy to learn long-term relationships. In addition, all neurons in the RNN layer are entangled, and their behavior is complex. Although the extended short-term memory network (LSTM) effectively solves these problems, using the hyperbolic tangent and Sigmoid functions will cause the gradient to decay

with the layer. Compared with RNN and LSTM neural networks, the neurons in the same layer of Bi-IndRNN are independent. The information between channels will spread and explore between multi-layer IndRNNs over time, and information can be cross-linked. Layer propagation improves the interpretability of neurons. At the same time, using unsaturated activation functions such as Relu solves the problem of gradient disappearance and explosion within and between layers, which enhances the robustness of the model and is excellent in processing long-term data more excellent. The Bi-IndRNN hidden layer state update at sampling time point $t$ is shown in the equation.

$$\begin{cases} \overrightarrow{h_t} = \sigma(\overrightarrow{W}\chi_{sub} + \overrightarrow{\mu} \odot \overrightarrow{h_{t-1}} + \overrightarrow{b}) \\ \overleftarrow{h_t} = \sigma(\overleftarrow{W}\chi_{sub} + \overleftarrow{\mu} \odot \overleftarrow{h_{t-1}} + \overleftarrow{b}) \end{cases} \tag{6}$$

where $\overrightarrow{h_t}$ and $\overleftarrow{h_t}$ represent the state update of the hidden layer in the positive and negative directions, respectively; $\odot$ represents the Hadamard product; $\overrightarrow{b}$ and $\overleftarrow{b}$ represent the bias value of the forward and reverse; $\overrightarrow{\mu}$ and $\overleftarrow{\mu}$ represent the weight of the forward and reverse, respectively. The state update of the hidden layer can obtain the initial features $f_{ir}$ of Bi-IndRNN, as shown in equation.

$$f_{ir} = \overrightarrow{h_t} + \overleftarrow{h_t} \tag{7}$$

Due to the differences in power at different sampling time points and the state of the device, at the same time, the feature semantics (global and context details) at different levels have different emphases. Fusing these different feature semantics with a simple concatenation strategy brings only slight performance improvements and increases the use of redundant information. To fully use the feature information at each level, the global semantics obtained by the dynamic graph convolution module, and the contextual semantic details obtained by Bi-IndRNN, we designed a local attention guidance layer to enrich their semantics. This operation does not sacrifice. In the case of any details, it makes up for the lack of semantic representation of a single feature and forms an effective interaction. Furthermore, when one of the features of one party is underrepresented, to avoid excessive interference with this feature, we add a residual design to emphasize the importance of the features of the other party. The local attention guidance layer is computed on the initial dynamic graph convolutional feature of $f_\varsigma$ and the Bi-IndRN feature of $f_{ir}$, as shown in equation.

$$\begin{cases} \alpha = \alpha_{att}(f_\varsigma, f_{ir}) \\ f_{ag} = f_{ir} + \alpha f_{ir} \end{cases} \tag{8}$$

where $f_{ag}$ denotes the initial features of the local attention guidance layer; $\alpha$ denotes the attention map; and $\alpha_{att}(\cdot)$ denotes the local attention operation. Furthermore, the calculation of the second local attention guidance layer is shown in the equation.

$$\begin{cases} \alpha' = \alpha'_{att}(f'_{ir}, f'_\varsigma) \\ f'_{ag} = f'_\varsigma + \alpha' f'_\varsigma \end{cases} \tag{9}$$

where $f'_{ag}$ denotes the final features of the local attention guidance layer; $\alpha'$ denotes the attention map; and $\alpha'_{att}(\cdot)$ denotes the local attention operation. Furthermore, the calculation of the feature $f'_{ir}$ is as follows:

$$\begin{cases} \overrightarrow{h'_t} = \sigma(\overrightarrow{W}f_{ag} + \overrightarrow{\mu} \odot \overrightarrow{h'_{t-1}} + \overrightarrow{b}) \\ \overleftarrow{h'_t} = \sigma(\overleftarrow{W}f_{ag} + \overleftarrow{\mu} \odot \overleftarrow{h'_{t-1}} + \overleftarrow{b}) \\ f'_{ir} = \overrightarrow{h'_t} + \overleftarrow{h'_t} \end{cases} \tag{10}$$

Subsequently, a superficial linear layer with a *Tanh* activation function is used to restore the desired output size, and the output value is multiplied by the maximum power of the device in the interval $[0, 1]$ to construct a reasonable energy prediction, and at the same time, by matching the corresponding threshold to obtain the device status, the specific calculation is shown in the equation.

$$O_{out} = Tanh(Liner(f'_{ag}))W' + b' \tag{11}$$

where $W'$ and $b'$ represent weight and bias values, and $Liner(\cdot)$ represents linear operation.

*3.5. Loss Function*

To accurately predict the energy and simultaneously perform state classification, we design a weighted loss function that tunes and optimizes the network while minimizing the absolute error. Among them, $\tau_{MSE}$ can not only accelerate the convergence speed but also reduce the gradient of the loss function as the error decreases, which helps the prediction result to be more stable. However, it is vulnerable to unmeasured discrete devices in the NILM sequence data or ignores the effects of these unmeasured devices, which leads to problems such as fitting or extraction explosion in the network during training; thus, the $\tau_{KL}$ loss function is introduced to assist $\tau_{MSE}$ to adjust and optimize the model. The weighted loss function of $\tau_{total}$ is shown in the equation:

$$\begin{cases} \tau_{total} = \tau_{MSE} + \beta\tau_{KL} \\ \tau_{MSE} = \frac{1}{N_c}\sum_{c=1}^{N_c}(Y_c - O_{out}^c)^2 \end{cases} \tag{12}$$

where $\tau_{MSE}$ indicates the mean squared error (MSE) loss; and $\tau_{KL}$ indicates the KL divergence loss. $\beta$ indicates the weighted factor, and $\beta = 0.3$. $N_c$ indicates the numbers of total sample.

In summary, we train the AttG-BDGNets method to generate predicted labels to match the actual labels and use a weighted loss function to optimize the proposed method. At the same time, the AdamW optimizer backpropagation is used to update the dynamic graph convolution module (DynamicGCM) weight and bias parameters of the Attention Guidance Module (AGM); the following Algorithm 1 demonstrates the representation learning and training process of the proposed AttG-BDGNets method on NILM-related data.

---

**Algorithm 1:** Training of AttG-BDGNets-based NILM-related data

---

**Input:** In the training set $\mathbb{D}$, the sequence feature matrix input by AttG-BDGNets is $\chi_{sub}$; the adjacency matrix $A_{mn} = (v_{m,n})_{N \times N}$ of DynamicGCM, the dynamic graph features $f_\varsigma$ and $f'_\varsigma$; the output features $f_{ag}$ and $f'_{ag}$ of AGM. Weighted loss function $\tau_{total}$ and optimizer of *AdamW*.

**for** $\iota = 0$ *to* $\iota^{Max}$ **do**

   **if** $\iota = 0$ **then**

      $f_\varsigma \leftarrow \varsigma(A, \chi_{sub})$ ;

      $f_{ir} \leftarrow (\overrightarrow{h_t}, \chi_{sub}) + (\overleftarrow{h_t}, \chi_{sub})$ ;

      $f_{ag} \leftarrow \alpha_{att}(f_{ir}, f_\varsigma)$

   **else**

      $f'_\varsigma \leftarrow f_\varsigma + \lambda f_\varsigma$ ;

      $f'_{ag} \leftarrow f'_\varsigma + \alpha_{att}(f'_\varsigma, f'_{ir})$

   **end**

   The output of AttG-BDGNets is calculation as $O_{out}$ according to Equation (11);

**end**

**output:** optimization the training during by weighted loss $\tau_{Total}$ and *AdamW*, according to Equation (12).

---

## 4. Experimental Results

To evaluate the effectiveness of the proposed AttG-BDGNets approach, we demonstrate it on open source baseline datasets, namely REDD and UK-DALE. This section briefly explains the dataset's source and provides the experimental results and detailed analysis.

### 4.1. Data Preparation

**REDD.** This datum measures the electricity usage of six residences in the United States, the total energy consumption data within a sampling period of 1 s, and the energy consumption data of household appliances within a sampling period of 3 s. It includes power data and device-specific usage of the main channel. It uses data to train a variety of specific appliances such as refrigerators, washers and dryers, microwave ovens, and dishwashers, one of which is used for model testing and remaining training [20].

**UK-Dale.** This dataset is mainly a household appliance level power dataset, which contains the power consumption level of five households in the UK for more than two years. Similarly, we focus on data from refrigerators, washing machines and dryers, microwave ovens, and dishwashers. We call a specific electrical appliance, with kettles as an additional target, unmetered or noise-item devices. Among them, two are used as testing samples, and the remaining ones are used as training samples [19].

### 4.2. Evaluation Index

To evaluate our proposed AttG-BDGNets method, the mean absolute error (MAE) and F-score are used as evaluation metrics. These evaluation metrics are calculated as shown in the equation.

$$\begin{cases} MAE = \frac{1}{M+K} \sum_{i=1}^{M+K} |Y_{pre}^i - Y_{gt}^i| \\ F_1 = \frac{2TP}{TP+2FP+FN)} \end{cases} \tag{13}$$

where $Y_{pre}^i, Y_{gt}^i$ represent the predicted power and absolute power of the $i^{th}$ device at the sampling time point $t$, respectively; the F-score value is designed to measure the accuracy of each device in the on/off state, and $TP$, $FP$, and $FN$, respectively, denote true positives, false positives, and false negatives. At the same time, when the adequate power of a specific device is greater than the set threshold, we consider the device to be in the "*On*" state, and when the active power is less than or equal to the set threshold, the device is considered to be in the "*Off*" state.

### 4.3. Parameter Settings

In the AttG-BDGNets method, the number of dynamic graph convolutional layers is set to 2, and the number of Bi-IndRNN layers is set to 4; that is, the initial feature extraction block contains two layers of Bi-InRNN. The number of local attention guidance layers is set to two; to ensure that the proposed AttG-BDGNets method obtains better prediction and classification performance, AdamW is used to optimize and adjust the network, and the initial learning rate is set to $1 \times 10^{-3}$, whilst the cosine annealing algorithm (CosineAnnealingWarmRestarts) is used to dynamically adjust the learning rate and set the code loss rate to 0.15, the batch processing to 128, and the number of iterations to 100. In addition, to ensure the smooth progress of the experiment, participate in the experiment. All methods are written on *ubuntu*18 using *Python*3.7.6, and the deep learning library includes *Numpy* and *Pytorch*1.7.0 + *cu*110, etc. At the same time, training and testing are completed on two *RTXA*5000 GPU graphics cards.

### 4.4. Ablation Study

To verify whether each module in the proposed NILM method of AttG-BDGNets has a positive effect on the entire network's performance, we evaluated and demonstrated each component on two open source NILM baseline data, namely REDD and UK-Dale. Furthermore, the corresponding experimental results and analysis are provided to prove

the effectiveness of each component. Table 1 presents the prediction and classification performance of the different components.

**Table 1.** Experimental results of different components. "DynamicGCM" and "Bi-IndRNN" indicate that only this module is used in the proposed AttG-BDGNets method; "ACL+RNN" indicates that we replace Bi-IndRNN with RNN; $\tau_{MSE}$ and $\tau_{KL}$ indicate that we use this loss function instead of the weighted loss we designed; "−" indicates REDD baseline data. There are no "Kettle" samples. Bold words indicate the best prediction and classification performances.

| Model Dataset | | REDD | | UK-Dale | |
| --- | --- | --- | --- | --- | --- |
| | | *MAE* | $F_1$ | *MAE* | $F_1$ |
| DynamicGCM | Fridge | 23.09 | 0.798 | 22.56 | 0.794 |
| | Washer | 25.62 | 0.727 | 8.72 | 0.508 |
| | Microwave | 16.02 | 0.561 | 7.09 | 0.593 |
| | Dishwasher | 15.67 | 0.608 | 12.54 | 0.665 |
| | Kettle | − | − | 8.14 | 0.943 |
| | Overall average | 20.2 | 0.677 | 11.81 | 0.701 |
| Bi-IndRNN | Fridge | 23.87 | 0.781 | 23.59 | 0.779 |
| | Washer | 26.02 | 0.715 | 9.33 | 0.486 |
| | Microwave | 16.88 | 0.547 | 8.12 | 0.584 |
| | Dishwasher | 17.03 | 0.584 | 13.78 | 0.651 |
| | Kettle | − | − | 10.08 | 0.925 |
| | Overall average | 20.95 | 0.657 | 12.98 | 0.685 |
| AGL+RNN | Fridge | 22.52 | 0.804 | 21.55 | 0.801 |
| | Washer | 24.16 | 0.734 | 8.31 | 0.511 |
| | Microwave | 15.39 | 0.571 | 6.83 | 0.599 |
| | Dishwasher | 15.22 | 0.619 | 12.04 | 0.67 |
| | Kettle | − | − | 7.44 | 0.947 |
| | Overall average | 19.32 | 0.682 | 11.23 | 0.706 |
| $\tau_{MSE}$ | Fridge | 21.73 | 0.804 | 21.36 | 0.804 |
| | Washer | 23.91 | 0.738 | 7.54 | 0.517 |
| | Microwave | 14.72 | 0.574 | 5.44 | 0.603 |
| | Dishwasher | 14.58 | 0.622 | 11.76 | 0.674 |
| | Kettle | − | − | 6.81 | 0.95 |
| | Overall average | 18.73 | 0.684 | 10.52 | 0.709 |
| $\tau_{KL}$ | Fridge | 21.39 | 0.801 | 22.6 | 0.805 |
| | Washer | 23.48 | 0.744 | 7.59 | 0.514 |
| | Microwave | 14.88 | 0.571 | 5.49 | 0.603 |
| | Dishwasher | 14.37 | 0.624 | 11.46 | 0.677 |
| | Kettle | − | − | 6.14 | 0.961 |
| | Overall average | 18.53 | 0.685 | 10.65 | 0.712 |
| AttG-BDGNets | Fridge | 21.18 | 0.812 | 20.36 | 0.808 |
| | Washer | 23.54 | 0.748 | 7.32 | 0.522 |
| | Microwave | 14.39 | 0.579 | 5.18 | 0.606 |
| | Dishwasher | 14.32 | 0.624 | 11.38 | 0.678 |
| | Kettle | − | − | 6.25 | 0.952 |
| | Overall average | **18.35** | **0.691** | **10.09** | **0.713** |

From Table 1, we can draw the following conclusions:

(1) In the NILM method of the proposed AttG-BDGNets, all components play a positive role in the model's overall performance; in particular, they cooperate in obtaining an optimal performance. For example, on the UK-Dale baseline dataset, the average F1 value of the proposed method using the weighted loss function of $\tau_{total}$ optimization is 0.001 and 0.004 higher than that of $\tau_{KL}$ and $\tau_{MSE}$, respectively, and the MAE is reduced by 0.56

and 0.43. The possible reason is that the weighted loss function and different loss functions are used to optimize DynamicGCM and AGM, respectively, reducing the absolute error between them. It is worth noting that the $\tau_{KL}$ loss function achieves the best performance on the Kettle sample of the UK-Dale sequence data. For example, the MAE and F1 values are, respectively, 0.11 lower and 0.09 higher than the NILM method of AttG-BDGNets. The proposed state-BDGNets and the NILM method obtained the best performances among all REDD devices.

(2) Compared with the Bi-IndRNN module, DynamicGCM has shown excellent prediction and classification capabilities on both data types. For example, on the REDD baseline data, the average MAE and F1 decreased by 0.93 and increased by 0.02, respectively—especially on the Dishwasher equipment, where the MAE decreased by 1.36 and the F1 value increased by 0.024. There are two possible reasons for this. On the one hand, DynamicGCM uses the node aggregation and transfer function in the layer, the detailed information is obtained to the maximum extent, and the repeated use of redundant noise is avoided. On the other hand, DynamicGCM realizes feature reuse through dynamic aggregation between layers, further avoiding details regarding the loss of information whilst efficient long-distance dependencies are established and applicable global semantics are captured, enhancing node representations and enabling the network to better track power device usage. Furthermore, this component is also demonstrated to improve the prediction and classification performance of e AttG-BDGNets method.

(3) AGL+RNN also showed strong competitiveness on the two baseline datasets. For example, the average F1 values of REDD and UK-Dale are 0.682 and 0.706, respectively. Compared with Bi-IndRNN, and there are differences in long-distance modeling and neuron interpretability—namely that modeling better captures the context details. In particular, Bi-IndRNN can better solve the problem of gradient disappearance and the explosion problem during the training phase and prevent the network from falling into an optimal local state. It is worth noting that the NILM method of the proposed AttG-BDGNets is more suitable for predicting and classifying devices with more operations, such as washing machines and dishwashers; it is less sensitive to the consumption signals of devices with fewer operations, such as kettles, microwave ovens, and refrigerators. Alternatively, it may be that the activation time of these devices is short and the operating state is lesser, so the proposed method cannot effectively predict the state of the device. In addition, to intuitively demonstrate that the weighted loss function can promote the proposed AttG-BDGNets method to learn better, Figure 2 shows the effect of different loss functions in the first 100 iterations of the model.

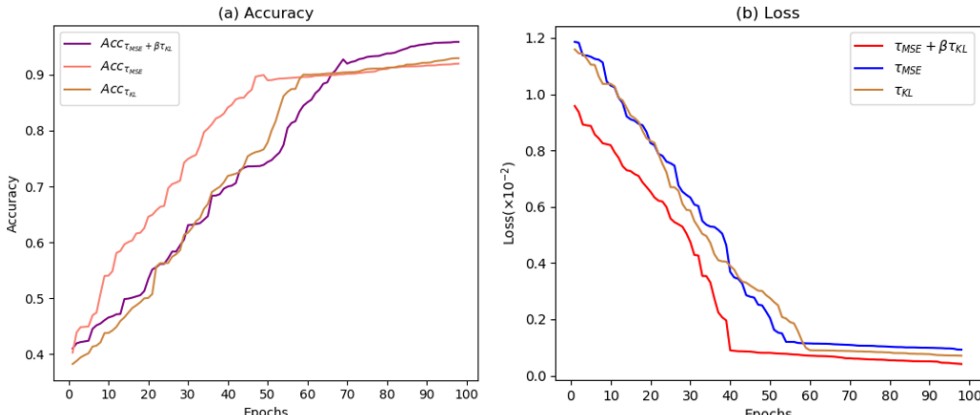

**Figure 2.** Different loss functions in the first 100 iterations of our models: (**a**) denotes the accuracy of different loss functions; and (**b**) represents the loss convergence of different loss functions. $\tau_{MSE} + \beta\tau_{KL}$ indicates the weighted loss function of our proposed, and $Acc_{\tau_{MSE}+\beta\tau_{KL}}$ indicates the accuracy of train.

Figure 2 shows that, under the same learning rate, the weighted loss function $\tau_{MSE} + \beta\tau_{KL}$ we designed converges relatively quickly, the decline is stable and relatively smooth, and the loss and accuracy tend to be stable after the 70th round. Compared with the weighted loss function $\tau_{MSE} + \beta\tau_{KL}$, we designed the convergence speed of the $\tau_{MSE}$ and $\tau_{KL}$ loss functions are poor. At the same time, the loss functions $\tau_{KL}$ and $\tau_{MSE}$ have multiple short-term intersections, and the loss ($\tau_{MSE}, \tau_{KL}$) and accuracy ($Acc_{\tau_{MSE}}, Acc_{\tau_{KL}}$) tend to be stable after 60 rounds, but under the same conditions, $\tau_{MSE}$ converges faster than $\tau_{KL}$. None of these loss functions oscillate too much; there is no severe fitting phenomenon.

### 4.5. Comparison with Other NILM Methods

To demonstrate the effectiveness and advancement of the proposed AttG-BDGNets method, we verified all methods on two open source NILM baseline data, namely REDD and UK-Dale, and gave the corresponding analysis. The experimental results of different NILM methods are shown in Table 2.

**Table 2.** Experimental results of different NILM methods.

| Model Dataset | | REDD | | UK-Dale | |
|---|---|---|---|---|---|
| | | $MAE$ | $F_1$ | $MAE$ | $F_1$ |
| DAE | Fridge | 30.14 | 0.735 | 27.94 | 0.658 |
| | Washer | 28.59 | 0.423 | 15.07 | 0.318 |
| | Microwave | 25.02 | 0.261 | 14.67 | 0.349 |
| | Dishwasher | 28.88 | 0.469 | 23.19 | 0.525 |
| | Kettle | – | – | 12.33 | 0.896 |
| | Overall average | 28.16 | 0.482 | 18.64 | 0.549 |
| LSTM | Fridge | 44.28 | 0.699 | 43.97 | 0.221 |
| | Washer | 36.28 | 0.215 | 18.04 | 0.350 |
| | Microwave | 19.35 | 0.577 | 9.02 | 0.384 |
| | Dishwasher | 27.33 | 0.424 | 39.98 | 0.601 |
| | Kettle | – | – | 20.14 | 0.827 |
| | Overall average | 31.81 | 0.479 | 26.23 | 0.477 |
| BERT | Fridge | 32.42 | 0.736 | 27.59 | 0.761 |
| | Washer | 35.72 | 0.538 | 8.98 | 0.467 |
| | Microwave | 18.89 | 0.502 | 7.83 | 0.289 |
| | Dishwasher | 22.61 | 0.516 | 17.45 | 0.632 |
| | Kettle | – | – | 7.88 | 0.902 |
| | Overall average | 27.41 | 0.573 | 13.94 | 0.61 |
| CNN | Fridge | 38.19 | 0.634 | 30.27 | 0.637 |
| | Washer | 38.37 | 0.257 | 14.38 | 0.259 |
| | Microwave | 20.16 | 0.429 | 8.95 | 0.357 |
| | Dishwasher | 26.18 | 0.509 | 28.04 | 0.537 |
| | Kettle | – | – | 11.21 | 0.789 |
| | Overall average | 30.72 | 0.457 | 18.57 | 0.516 |
| SGN | Fridge | 27.73 | 0.615 | 17.72 | 0.799 |
| | Washer | 30.48 | 0.654 | 12.97 | 0.593 |
| | Microwave | 18.81 | 0.437 | 8.26 | 0.519 |
| | Dishwasher | 17.74 | 0.538 | 12.11 | 0.526 |
| | Kettle | – | – | 10.1 | 0.923 |
| | Overall average | 23.69 | 0.561 | 12.23 | 0.672 |

**Table 2.** *Cont.*

| Model Dataset | | REDD | | UK-Dale | |
|---|---|---|---|---|---|
| | | *MAE* | *F₁* | *MAE* | *F₁* |
| LDwA | Fridge | 23.88 | 0.776 | 15.42 | **0.821** |
| | Washer | 25.97 | 0.72 | 9.65 | 0.664 |
| | Microwave | **13.14** | **0.658** | 5.69 | 0.575 |
| | Dishwasher | **10.56** | **0.714** | 9.77 | 0.627 |
| | Kettle | – | – | 7.68 | **0.981** |
| | Overall average | 18.38 | 0.717 | **9.64** | **0.733** |
| GCN | Fridge | **17.66** | **0.858** | 30.17 | 0.194 |
| | Washer | 30.09 | 0.668 | 12.35 | 0.485 |
| | Microwave | 20.65 | 0.421 | 8.77 | 0.268 |
| | Dishwasher | 10.38 | 0.584 | 41.96 | 0.562 |
| | Kettle | – | – | 18.33 | 0.908 |
| | Overall average | 19.69 | 0.632 | 22.31 | 0.483 |
| AttG-BDGNets | Fridge | 21.18 | 0.812 | 20.36 | 0.808 |
| | Washer | 23.54 | 0.748 | 7.32 | 0.522 |
| | Microwave | 14.39 | 0.579 | 5.18 | 0.606 |
| | Dishwasher | 14.32 | 0.624 | 11.38 | 0.678 |
| | Kettle | – | – | 6.25 | 0.952 |
| | Overall average | **18.35** | **0.691** | **10.09** | **0.713** |

From the experimental results in Table 2, we found that:

(1) On baseline datasets REDD and UK-Dale, the average result of the AttG-BDGNets method is better than other NILM methods and has a tremendous competitive advantage over most devices, but devices that handle fewer operations exist. There are disadvantages, however, including the MAE of the refrigerator in the REDD data being 3.52 higher than that of GCN, and the F1 value being 0.046 lower than that of GCN; it may be possible to further enhance the performance of rarely used devices by expanding the training sample size and improving the topology map construction strategy. On other devices, there may be two reasons for the better performance of our proposed method. First, the global features acquired by DynamicGCM and the context details extracted by AGM form a complementary relationship, enhancing the features' representation and making the method more efficient. Paying attention to the state of less-operated equipment, secondly, the designed weighted loss function, by updating network parameters to keep detailed information in time, improves the performance of less commonly used electrical equipment, such as kettles. In addition, the proposed AttG-BDGNets method takes devices as graph nodes and all the attributes of each device as node features to construct the topology graph. The edge weights between them highlight the direct differences between different devices and improve the final performance.

(2) The two most competitive methods are LDwA and GCN, and their average MAEs on UK-Dale are 9.64 and 22.31, respectively. Among them, GCN also realizes prediction and classification by gathering node information—while LDwA is encoding. The decoding structure, CNN, is used to obtain local features, and LSTM models the global semantics, thus prompting the network to improve the performance of less-operated devices. Similarly, our method is better than SGN on most devices, which means that our method is very beneficial for establishing complementary features for representation. At the same time, it also shows that this local attention guidance method can significantly refine features while improving power consumption estimation and load classification, which improves the overall performance. In more detail, compared with SGN, CNN, and BERT for the dataset REDD, the refrigerator MAE of our proposed AttG-BDGNets is reduced by 6.55, 17.01, and 11.24, respectively. On the f1 score, the classification performance of the refrigerator improved by 0.179, 0.178, and 0.076, respectively. In addition, our method outperformed

DAE on all datasets, such as the dishwasher's F1 value, which improved by 0.155 and 0.153, respectively.

### 4.6. Discussion

Regardless of whether the designed NILM method is effective, the model complexity is an important indicator. The NILM method of AttG-BDGNets proposed by us still has a substantial competitive advantage in terms of time efficiency while obtaining the best prediction and classification accuracy. Figure 3 shows the operating efficiency and parameter quantity of different NILM methods. Among them, FLOPS represents the number of floating-point operations per second, and the unit is GB; "Parm" represents the number of model parameters; it is worth noting that the larger the value of FLOPS, the higher the performance of the model.

Figure 4 demonstrates the power prediction of different electrical equipment using the NILM method of our proposed AttG-BDGNets on the REDD and UK-Dale datasets. The sampling time predicted in REDD is May 2011 (instances include fridge, microwave, and light). The predicted sampling time points in the UK-Dale data are from January to April 2014 (examples include fridge, kettle, and light).

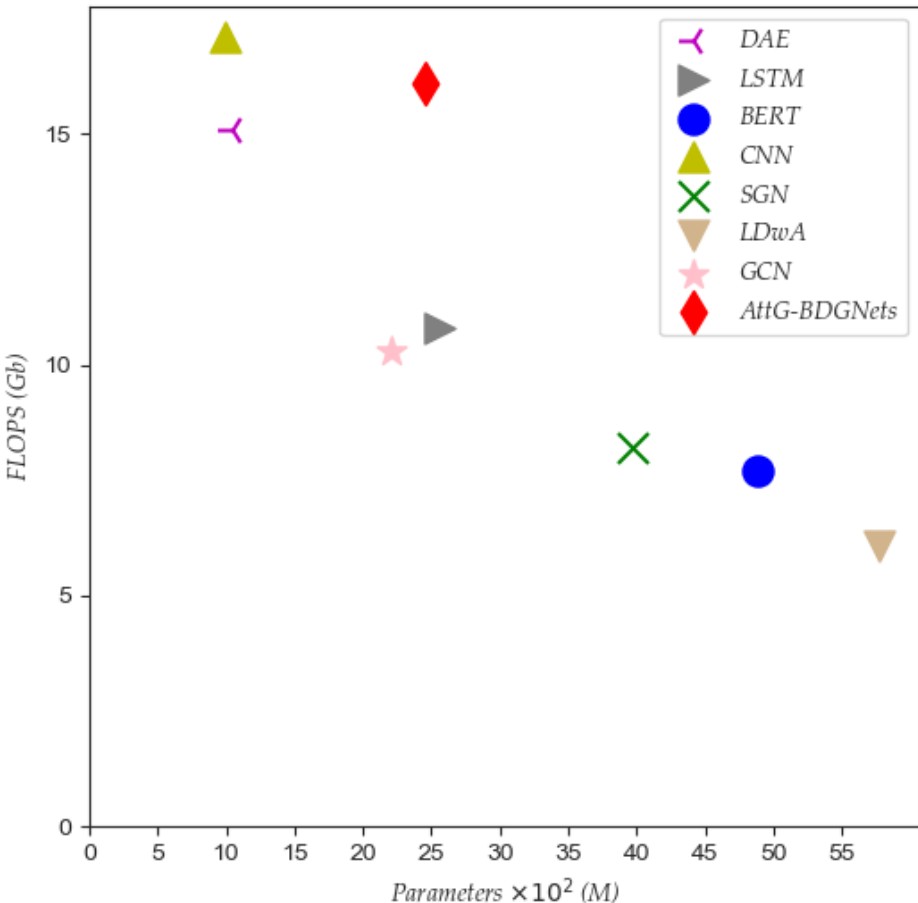

**Figure 3.** The operating efficiency and parameter quantity of different NILM methods.

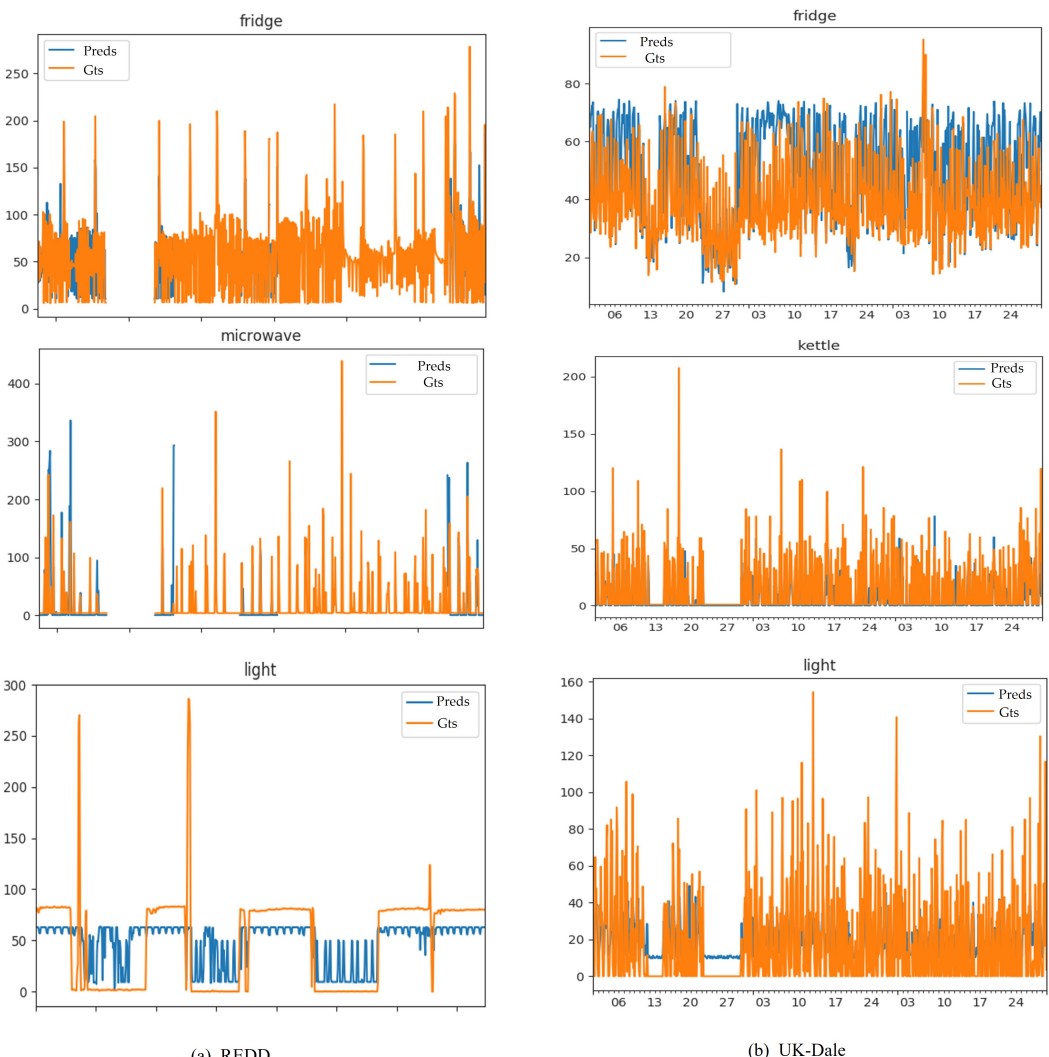

<div align="center">(a) REDD</div>

<div align="center">(b) UK-Dale</div>

**Figure 4.** Visual presentation of our models: (**a**) indicates the REDD dataset; and (**b**) indicates the UK-Dale dataset. The blue line of 'preds' represents the predicted value of our proposed method, and the yellow line of 'Gts' represents the ground truth.

## 5. Conclusions and Future Research

This paper designs a new attention-guided bidirectional dynamic graph IndRNN method (AttG-BDGNets) for NILM problems. This enables us to include the device-under-test and unmetered device-specific power usage. The bidirectional independent recurrent neural network promotes the extraction of contextual features from both positive and negative directions. At the same time, the local attention guidance layer is used to integrate the dynamic graph's global features and contextual semantics to form a complementary relationship and promote the network to learn more compelling features. The method is evaluated and verified on two open source baseline datasets, namely REDD and UK-DALE. The final experimental results show that our proposed method is better than other popular NILM methods in terms of all devices' load recognition accuracy and prediction ability.

During the construction and training of the topology map, we found two shortcomings of this method. First, the structure of the topology map directly affects the load recognition accuracy; that is to say, the better the topology map structure, the higher the recognition accuracy. Second, compared with the straightforward NILM method, the execution efficiency of our proposed method needs to be improved. In the following research, we will start from these two points and develop a faster and more concise dynamic semantic graph

model to reduce the complexity of the model and further improve the load identification accuracy and prediction ability.

**Author Contributions:** Methodology, Z.W.; software, Z.W.; resources, X.Z.; writing—original draft, Z.W.; writing—review and editing, X.Z.; visualization, Z.W.; supervision, X.Z.; project administration, X.Z. All authors have read and agreed to the published version of the manuscript.

**Funding:** This research is supported by the Fundamental Research Funds for the Central Universities (No.: 2020ZDPY0223).

**Informed Consent Statement:** All participating authors agree to publish.

**Acknowledgments:** I am very grateful to my peers for their support and to the reviewers who gave constructive comments.

**Conflicts of Interest:** The authors declare no conflict of interest.

## Abbreviations

The following abbreviations are used in this manuscript:

| | |
|---|---|
| AttG-BDGNets | Attention-guided bidirectional dynamic graph IndRNN method |
| NILM | Non-intrusive load monitoring |
| IndRNN | Independently recurrent neural network |
| AGM | Attention guidance module |
| DynamicGCM | Dynamic graph convolution |
| Bi-IndRNN | Bidirectional independent recurrent neural network |
| GCN | Graph convolution network |

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
