# Peer review of "AttG-BDGNets: Attention-Guided Bidirectional Dynamic Graph IndRNN for Non-Intrusive Load Monitoring"

_information, doi:10.3390/info14070383_

Round 1

Reviewer 1 Report

The following comments must be revised before acceptance for publication.

(1) The specific structure of AttG-BDGNets should be introduced in detail.

(2) Why Bi-IndRNN and local attention are adopted in this work, the motivation is their good learning property or high efficiency?

(3) Since the convergence of depth learning is still challenging, the convergence curve of the objective function of the deep learning model is encouraged to be presented.

(4) More samples should be displayed to help understand the difficulty of the task.

(5) The summary and description of related work in the field are insufficient. The following related work of deep learning must be cited and discussed, including “VoxelEmbed: 3D instance segmentation and tracking with voxel embedding based deep learning.” International Workshop on Machine Learning in Medical Imaging. Springer, Cham, 2021: 437-446. “Compound figure separation of biomedical images with side loss.” Deep Generative Models, and Data Augmentation, Labelling, and Imperfections. Springer, Cham, 2021. 173-183. “Improvement of generalization ability of deep CNN via implicit regularization in two-stage training process,” IEEE Access, vol. 6, pp. 15844-15869, 2018.

Author Response

(1) The specific structure of AttG-BDGNets should be introduced in detail.

Response (1): Thanks a lot for the advice from the experts; we have revised it in detail.

The overall network structure of AttG-BDGNets is shown in Figure 1. The proposed AttG-BDGNets monitoring method consists of two essential modules: the dynamic graph convolution module (DynamicGCM) and the attention guidance module (AGM). The dynamic graph convolution module (DynamicGCM) aims to gather and transmit functions to obtain the hidden dynamic information in the NILM sequence. Simply put, the load actions included in the NILM sequence have good steady-state and transient performance; that is, their transition process should be taken into account when obtaining transient characteristics and the instantaneous Power and switching transient characteristics, so the extraction process of steady-state features is more accessible to obtain than transient features. The dynamic graph convolution module (DynamicGCM) can capture this dynamic information and use the dynamics of the inter-layer neighborhood. The aggregation strategy learns node representations from the graph and directly associates distant correlation graph nodes to effectively learn longer temporal dependencies, forming an effective interaction between steady-state features and transient features. AGM mainly includes a local attention guidance layer and a bidirectional independent recurrent neural network module, in which the bidirectional independent recurrent neural network (Bi-IndRNN) models the power consumption data sequence in different periods from the positive and negative directions and obtains Contextual semantic details, and fully capture the correlation between changes in electricity usage data and household members' electricity usage behavior. The Local Attention Guidance Layer (LAG) combines the global features captured by the Dynamic Graph Convolution Module (DynamicGCM) with the contextual semantics acquired by the Bi-Independent Recurrent Neural Network (Bi-IndRNN), forming a complementary relationship between them to Improve the semantic representation ability, that is when the representation effect of  is not good, use this guide layer to highlight the representation of  and prevent  from affecting On the whole, we first process the input NILM sequence and input the initial dynamic graph convolution module (DynamicGCM) and bidirectional independent recurrent neural network (Bi-IndRNN), respectively, to obtain the corresponding feature semantics  and ,and use the local attention-guided layer performs feature fusion to strengthen semantic representation; finally, accurate prediction is achieved.

(2) Why Bi-IndRNN and local attention are adopted in this work, the motivation is their good learning property or high efficiency?

Response (2): Thanks a lot for the advice from the experts; we have revised it in detail.

AGM mainly consists of a bidirectional independent recurrent neural network (Bi-IndRNN) and a local attention guidance layer. Among them, Bi-IndRNN models the input NILM sequence from both positive and negative directions, aiming to explore the correlation between the total power and all devices and to construct the long-distance dependence between them in different periods. The traditional recurrent neural network (RNN) cannot effectively solve the problems of gradient disappearance and gradient explosion during the training process, and it isn't easy to learn long-term relationships. In addition, all neurons in the RNN layer are entangled, and their behavior is complex. Although the extended short-term memory network (LSTM) effectively solves these problems, using hyperbolic tangent and Sigmoid functions will cause the gradient to decay with the layer. Compared with RNN and LSTM neural networks, the neurons in the same layer of Bi-IndRNN are independent. The information between channels will spread and explore between multi-layer IndRNNs over time, and information can be cross-linked. Layer propagation improves the interpretability of neurons. At the same time, using unsaturated activation functions such as Relu solves the problem of gradient disappearance and explosion within and between layers, which enhances the robustness of the model and is excellent in processing long-term data more excellent.

Due to the differences in the power at different sampling time points and the state of the device, at the same time, the feature semantics (global and context details) at different levels have different emphases. Fusing these different feature semantics with a simple concatenation strategy brings only slight performance improvements and increases the use of redundant information. To fully use the feature information at each level, the global semantics obtained by the dynamic graph convolution module, and the contextual semantic details obtained by Bi-IndRNN, we designed a local attention guidance layer to enrich their semantics. This operation does not sacrifice. In the case of any details, it makes up for the lack of semantic representation of a single feature and forms an effective interaction. Furthermore, when one of the features of one party is underrepresented, to avoid excessive interference with this feature, we add a residual design to emphasize the importance of the features of the other party.

(3) Since the convergence of depth learning is still challenging, the convergence curve of the objective function of the deep learning model is encouraged to be presented.

Response (3):. Thanks a lot for the advice from the experts; we have revised it in detail.

Figure 2 shows that under the same learning rate, the weighted loss function   we designed converges relatively quickly, the decline is stable and relatively smooth, and the loss and accuracy tend to be stable after the 70 rounds. Compared with the weighted loss function   we designed the convergence speed of the   and loss functions is poor. At the same time, the loss functions and have multiple short-term intersections, and the loss    and accuracytend to be stable after 60  rounds, but Under the same conditions, converges faster than None of these loss functions oscillates too much; there is no severe fitting phenomenon.

(4) More samples should be displayed to help understand the difficulty of the task.

Response (4):. Thanks a lot for the advice from the experts; we have revised it in detail.

Figure 4 demonstrates the power prediction of different electrical equipment using the NILM method of our proposed AttG-BDGNets on the REDD and UK-Dale datasets. The sampling time predicted in REDD is May 2011 (instances include fridge, microwave, and light). The predicted sampling time points in the UK-Dale data are from January to April 2014 (examples include fridge, kettle, and light).

(5) The summary and description of related work in the field are insufficient. The following related work of deep learning must be cited and discussed, including “VoxelEmbed: 3D instance segmentation and tracking with voxel embedding based deep learning.” International Workshop on Machine Learning in Medical Imaging. Springer, Cham, 2021: 437-446. “Compound figure separation of biomedical images with side loss.” Deep Generative Models, and Data Augmentation, Labelling, and Imperfections. Springer, Cham, 2021. 173-183. “Improvement of generalization ability of deep CNN via implicit regularization in two-stage training process,” IEEE Access, vol. 6, pp. 15844-15869, 2018.
Response (5):. Thanks a lot for the advice from the experts; we have revised it in detail.

In recent years, deep learning methods have received extensive attention in many fields, such as medical image generation and detection, relying on their strong self-learning ability[15][16][17][18]. Many researchers have been inspired to apply these techniques to NILM tasks.  For example, Kelly et al. [19] applied deep neural networks to non-intrusive load monitoring tasks and achieved accurate predictions, then used memory networks such as LSTM to process high-frequency Time series to predict the start time, end time, and average power demand of each device. Considering the problems of explosive gradients, vanishing gradients, and network degradation in deep neural networks, Jia Z et al. [20] proposed a non-intrusive load monitoring method based on bidirectional extended convolution for learning low-frequency data, mainly relying on temporal convolution.

[15] Zhao M, Liu Q, Jha A, et al. VoxelEmbed: 3D instance segmentation and tracking with voxel embedding based deep learning[C]//Machine Learning in Medical Imaging: 12th International Workshop, MLMI 2021, Held in Conjunction with MICCAI 2021, Strasbourg, France, September 27, 2021, Proceedings 12. Springer International Publishing, 2021: 437-446.

[16] Yao T, Qu C, Liu Q, et al. Compound figure separation of biomedical images with side loss[C]//Deep Generative Models, and Data Augmentation, Labelling, and Imperfections: First Workshop, DGM4MICCAI 2021, and First Workshop, DALI 2021, Held in Conjunction with MICCAI 2021, Strasbourg, France, October 1, 2021, Proceedings 1. Springer International Publishing, 2021: 173-183.

[17] Yao T, Qu C, Liu Q, et al. Deep Generative Models, and Data Augmentation, Labelling, and Imperfections[J]. 2021.

[18] Zheng Q, Yang M, Yang J, et al. Improvement of generalization ability of deep CNN via implicit regularization in two-stage training process[J]. IEEE Access, 2018, 6: 15844-15869.

Reviewer 2 Report

The author introduced the attention-based dynamic graph IndRNN technique to tackle non-intrusive load monitoring issues, and then evaluated and verified the algorithm using two publicly available datasets, namely REDD and UK-DALE. According to the outcomes, the proposed algorithm surpassed other commonly used approaches in terms of both load recognition accuracy and prediction ability. Although the algorithm is valuable and worthy of publication, the quality of the English language and presentation requires extensive editing.

The detailed comments are as follows:

p2, line53: what is the meaning of the word "Forget." here?

p2, line56: what is the meaning of the sentence "Mining, resutling in poor accuracy of monitoring or decomposition models and suboptimal generalization performance"?

p4, line188: what T means?

p5, line200-204: why the extraction process of stea-state features is more accessible to obtain that transient features? Is there any reference that describes the usage of DynamicGCM that support your argument in this paragraph? If any, please include the reference here.

p5, line236, what is the meaning of "Eq."?

The description of variables ought to be included in the main body of the text rather than in the caption of Figure 1.

p7, line266, line278, line283: what is the meaning of "Eq."?

p9, Section 4.1: This section, Data Preparation, should contain specifics regarding the dataset, such as the total number of data points, the proportion of training and testing datasets, and the method for selecting testing samples.

In Fig2, The caption states that the first 50 iterations are being shown, yet the x-axis of the graph only displays 20 epochs. Furthermore, the trends of the three curves have yet to converge, suggesting that the number of epochs displayed may not be sufficient to observe convergence. Is there a plausible explanation for this discrepancy?

The extensive discussions for Fig3 and Fig4 are required. 

Author Response

The author introduced the attention-based dynamic graph IndRNN technique to tackle non-intrusive load monitoring issues, and then evaluated and verified the algorithm using two publicly available datasets, namely REDD and UK-DALE. According to the outcomes, the proposed algorithm surpassed other commonly used approaches in terms of both load recognition accuracy and prediction ability. Although the algorithm is valuable and worthy of publication, the quality of the English language and presentation requires extensive editing.

The detailed comments are as follows:

p2, line53: what is the meaning of the word "Forget." here?

Response (1):. Thanks a lot for the advice from the experts; we have revised it in detail.

That is, they can learn long-term distance information on sequence data. At the same time, to avoid redundant information being reused, forgetting gates are used to achieve memory selection.

p2, line56: what is the meaning of the sentence "Mining, resutling in poor accuracy of monitoring or decomposition models and suboptimal generalization performance"?

Response (2):. Thanks a lot for the advice from the experts; we have revised it in detail.

Although these methods maximize information and reduce errors caused by manual participation, capturing details from signals in non-intrusive load monitoring (NILM) is difficult. There is a lack of sufficient knowledge of users' electricity consumption habits. Resulting in poor accuracy of monitoring or decomposition models and suboptimal generalization performance. To address these issues, we propose an attention-oriented bidirectional dynamic graph IndRNN method, which aggregates and transfers node information by building a topological graph, and extracts device-specific power usage from aggregated signals, aiming at accurate non- Intrusive Load Monitoring (NILM).

p4, line188: what T means?

Response (3):. Thanks a lot for the advice from the experts; we have revised it in detail.

Non-intrusive load monitoring (NILM) was first proposed by George W in 1992. Assuming that the total power consumption of all devices is , there are metered devices under test and unmetered devices,   is the sampling time , namely, indicates the length of input sequence;

p5, line200-204: why the extraction process of stea-state features is more accessible to obtain that transient features? Is there any reference that describes the usage of DynamicGCM that support your argument in this paragraph? If any, please include the reference here.

Response (4):. Thanks a lot for the advice from the experts; we have revised it in detail.

The overall network structure of AttG-BDGNets is shown in Figure 1. The proposed AttG-BDGNets monitoring method consists of two essential modules: the dynamic graph convolution module (DynamicGCM) and the attention guidance module (AGM). The dynamic graph convolution module (DynamicGCM) aims to gather and transmit functions to obtain the hidden dynamic information in the NILM sequence. Simply put, the load actions included in the NILM sequence have good steady-state and transient performance; that is, their transition process should be taken into account when obtaining transient characteristics and the instantaneous Power and switching transient characteristics, so the extraction process of steady-state features is more accessible to obtain than transient features. The dynamic graph convolution module (DynamicGCM) can capture this dynamic information and use the dynamics of the inter-layer neighborhood. The aggregation strategy learns node representations from the graph and directly associates distant correlation graph nodes to effectively learn longer temporal dependencies, forming an effective interaction between steady-state features and transient features. AGM mainly includes a local attention guidance layer and a bidirectional independent recurrent neural network module, in which the bidirectional independent recurrent neural network (Bi-IndRNN) models the power consumption data sequence in different periods from the positive and negative directions and obtains Contextual semantic details, and fully capture the correlation between changes in electricity usage data and household members' electricity usage behavior. The Local Attention Guidance Layer (LAG) combines the global features captured by the Dynamic Graph Convolution Module (DynamicGCM) with the contextual semantics acquired by the Bi-Independent Recurrent Neural Network (Bi-IndRNN), forming a complementary relationship between them to Improve the semantic representation ability.

Faustine A et al. [28] considered that the performance of the device classifier is highly dependent on the signal characteristics of the load, but it is difficult to effectively distinguish between similar signals, which increases the difficulty of subsequent device classification, and proposed a weighted cycle graph (WRG) for Representing these signals improves the performance of device classification. Jiao X et al. [29] considered that many existing research methods have specific load decomposition errors when dealing with multi-modal devices. A non-intrusive load monitoring model based on a graph neural network is proposed. The graph structure is used to represent the relevant information between data nodes, combined with the long-term and short-term memory to extract the data's time-domain characteristics while retaining the power data's time characteristics. The correlation between different modes of equipment is improved, and the decomposition error is reduced. Jie Y et al. [30] proposed a non-intrusive load decomposition model based on a graph convolutional network (GCN) by fully mining the user's electricity consumption habits. The model first uses the time features extracted from the user's electricity consumption habits and constructs the power sequence as graph data based on the spectral graph theory as network input; then, based on the graph convolutional neural network, extracts the power attribute characteristics of each electrical appliance and its correlation Time-dependent structural features for non-intrusive load decomposition.

[28] Faustine A, Pereira L. Improved appliance classification in non-intrusive load monitoring using weighted recurrence graph and convolutional neural networks[J]. Energies, 2020, 13(13): 3374.

[29] Jiao X, Chen G, Liu J. A Non-Intrusive Load Monitoring Model Based on Graph Neural Networks[C]//2023 IEEE 2nd International Conference on Electrical Engineering, Big Data and Algorithms (EEBDA). IEEE, 2023: 245-250.

[30] Jie Y, Yajuan Q, Lihui W, et al. Non-Intrusive Load Decomposition Based on Graph Convolutional Network[C]//2022 IEEE 5th International Electrical and Energy Conference (CIEEC). IEEE, 2022: 1941-1944.

p5, line236, what is the meaning of "Eq."?

Response (5):. Thanks a lot for the advice from the experts; we have revised it in detail.

Assume that the metered device under test is , the unmetered device is , the total power sequence of all devices is  ,is the sampling time, the sliding window length is , and the  power sequence is . Secondly, the sliding window is moved by  sampling time points. Finally, power sequences are obtained so that each power sequence can be represented as a topological graph, where there are   nodes in the graph, that is, devices contribute to each power sequence, nodes There is a correlation between them, and this correlation is called the edge weight between nodes.  To put it simply, we build a topology map with power sequences. Each power sequence has several electrical devices that generate power in different periods, and each device is a node of the topology graph. Formally, the topological graph  is shown in Eq (2).

(2)

The description of variables ought to be included in the main body of the text rather than in the caption of Figure 1.

Response (6):. Thanks a lot for the advice from the experts; we have revised it in detail.

p7, line266, line278, line283: what is the meaning of "Eq."?

Response (7):. Thanks a lot for the advice from the experts; we have revised it in detail.

AGM mainly consists of a bidirectional independent recurrent neural network (Bi-IndRNN) [32] and a local attention [33] guidance layer. Among them, Bi-IndRNN models the input NILM sequence from both positive and negative directions, aiming to explore the correlation between the total power and all devices and to construct the long-distance dependence between them in different periods. The traditional recurrent neural network (RNN) cannot effectively solve the problems of gradient disappearance and gradient explosion during the training process, and it isn't easy to learn long-term relationships. In addition, all neurons in the RNN layer are entangled, and their behavior is complex. Although the extended short-term memory network (LSTM) effectively solves these problems, using hyperbolic tangent and Sigmoid functions will cause the gradient to decay with the layer. Compared with RNN and LSTM neural networks, the neurons in the same layer of Bi-IndRNN are independent. The information between channels will spread and explore between multi-layer IndRNNs over time, and information can be cross-linked. Layer propagation improves the interpretability of neurons. At the same time, using unsaturated activation functions such as Relu solves the problem of gradient disappearance and explosion within and between layers, which enhances the robustness of the model and is excellent in processing long-term data more excellent. Bi-IndRNN hidden layer state update at sampling time point t is shown in equation (6).

(6)

p9, Section 4.1: This section, Data Preparation, should contain specifics regarding the dataset, such as the total number of data points, the proportion of training and testing datasets, and the method for selecting testing samples.

Response (8):. Thanks a lot for the advice from the experts; we have revised it in detail.

 REDD. [21] This data measures the electricity usage of 6 residences in the United States, the total energy consumption data within a sampling period of 1 second, and the energy consumption data of household appliances within a sampling period of $3$ seconds. It includes power data and device-specific usage of the main channel. It uses data to train a variety of specific appliances such as refrigerators, washers and dryers, microwave ovens, and dishwashers, of which 1 is used for model testing and remaining training.

 UK-Dale.[19] This data set is mainly a household appliance level power data set, which contains the power consumption level of 5 households in the UK for more than two years. Similarly, we focus on refrigerators, washing machines and dryers, microwave ovens, and dishwashers data. We call a specific electrical appliance, with kettles as an additional target, unmetered or noise-item devices. Among them, 2 are used as testing samples, and remaining are used as training samples.

In Fig2, The caption states that the first 50 iterations are being shown, yet the x-axis of the graph only displays 20 epochs. Furthermore, the trends of the three curves have yet to converge, suggesting that the number of epochs displayed may not be sufficient to observe convergence. Is there a plausible explanation for this discrepancy?

Response (9):. Thanks a lot for the advice from the experts; we have revised it in detail.

Figure 2 shows that under the same learning rate, the weighted loss function   we designed converges relatively quickly, the decline is stable and relatively smooth, and the loss and accuracy tend to be stable after the 70 rounds. Compared with the weighted loss function   we designed the convergence speed of the   and loss functions is poor. At the same time, the loss functions and have multiple short-term intersections, and the loss    and accuracytend to be stable after 60  rounds, but Under the same conditions, converges faster than None of these loss functions oscillates too much; there is no severe fitting phenomenon.

The extensive discussions for Fig3 and Fig4 are required. 

Response (10):. Thanks a lot for the advice from the experts; we have revised it in detail.

Whether or not the designed NILM method is effective, the model complexity is an important indicator. The NILM method of AttG-BDGNets proposed by us still has a substantial competitive advantage in terms of time efficiency while obtaining the best prediction and classification accuracy. Figure 3 shows the operating efficiency and parameter quantity of different NILM methods. Among them, FLOPS represents the number of floating-point operations per second, and the unit is GB; "Parm" represents the number of model parameters; it is worth noting that the larger the value of FLOPS, the higher the performance of the model.

Figure 4 demonstrates the power prediction of different electrical equipment using the NILM method of our proposed AttG-BDGNets on the REDD and UK-Dale datasets. The sampling time predicted in REDD is May 2011 (instances include fridge, microwave, and light). The predicted sampling time points in the UK-Dale data are from January to April 2014 (examples include fridge, kettle, and light).

Reviewer 3 Report

The authors propose a method for non-intrusive load monitoring based on graph neural networks. Specifically, they develop DynamicGCM, AGM, and their relationships, representing a commendable attempt to apply graph-based methodology to non-intrusive load monitoring. However, there are several areas for improvement that need to be addressed before publication.

1. the manuscript contains several typographical and grammatical errors that need to be carefully modified throughout.

2.  the introduction lacks a clear explanation as to why graph-based methods are more suitable than DNNs for non-intrusive load monitoring. The comparison presented is only with conventional ML.

3. it is unclear how the proposed method deals with temporal dependencies, particularly long-term ones. It is difficult to understand how GNNs identify and learn temporal dependencies or patterns.

4. the "Related works" section does not include any information on graph neural networks or their applications to non-intrusive load monitoring. It is recommended that the authors review GNN-based approaches in related fields.

5. even in Chapter 3, there is no clear definition or explanation of how the method deals with temporal information and dependency. It is unclear how GCN interprets the data samples produced from different time intervals. Does the model focus only on graph structures for finding optimal feature representations? The data flow and learning strategy of the model, particularly regarding time, are too ambiguous.

6. the choice to adopt both MSE and KL in the loss function design requires further explanation. There is a concern that the proposed loss function may result in high residual for the learning model. The authors should explain the justification for the proposed loss function.

Author Response

The authors propose a method for non-intrusive load monitoring based on graph neural networks. Specifically, they develop DynamicGCM, AGM, and their relationships, representing a commendable attempt to apply graph-based methodology to non-intrusive load monitoring. However, there are several areas for improvement that need to be addressed before publication.

  1. the manuscript contains several typographical and grammatical errors that need to be carefully modified throughout.

Response 1: Thanks a lot for the advice from the experts; we have revised it in detail.

Electricity has become an indispensable resource for human life with the increase in social population and the continuous progress of various industries. However, due to the reduction and depletion of fossil resources, how to effectively save electricity resources and avoid waste has been widely concerned. In the power saving process, power consumption management and energy efficiency optimization are essential; non-intrusive load monitoring (NILM) [1]  is used to estimate the power demand of each device in the case of monitoring the total power demand signal recorded by a single meter of multiple devices, to formulate corresponding energy-saving measures, that is, it is adequate to improve energy consumption management and energy efficiency analysis to achieve maximum cost savings. In addition, non-intrusive load detection can more accurately understand the composition of users' electricity consumption, help power companies to formulate electricity prices and allocate resources scientifically and rationally, and at the same time, provide a more accurate basis for planning and scheduling issues in the power system. In addition, non-intrusive load monitoring (NILM) can prompt electrical appliance users to intuitively understand the power consumption and operation of various equipment in different periods and regulate and adjust power consumption behavior.

In the early stage, to achieve accurate non-intrusive load monitoring (NILM), most of them used simple machine learning methods for optimization analysis, such as k-nearest neighbor (k-NN) [2], support vector machine (SVM) [3], matrix decomposition [4], etc.; these methods mainly implement power measurement by sampling at intervals of seconds or minutes. Hidden Markov models [5]  benefited from the conversion of consumption levels that simulated the actual energy consumption of the target equipment and were widely used in the early stages. Although the early methods were energy-saving, they required manual participation in feature setting and screening. Their accuracy often depended on human subjectivity, so they relied heavily on previous expert experience and were challenging to apply on a large scale. In recent years, the successful application of deep learning technology in many fields also provides a new idea for non-intrusive load monitoring (NILM). For example, the convolutional neural network [6] automatically learns practical details by changing the activation state of neurons, avoiding Human participation in feature selection, and the setting is widely used. Still, convolutional networks perform poorly in the establishment of long-term dependencies. Long short-term memory [7] and recurrent neural networks [8] have attracted attention because of their ability to learn long-term relationships; they can learn long-term distance information on sequence data. At the same time, to avoid redundant information being reused, forgetting gates are used to achieve memory selection. Although these methods maximize information and reduce errors caused by manual participation, capturing details from signals in non-intrusive load monitoring (NILM) is difficult. There is a lack of sufficient knowledge of users' electricity consumption habits. Resulting in poor accuracy of monitoring or decomposition models and suboptimal generalization performance. To address these issues, we propose an attention-oriented bidirectional dynamic graph IndRNN method, which aggregates and transfers node information by building a topological graph, and extracts device-specific power usage from aggregated signals, aiming at accurate non- Intrusive Load Monitoring (NILM).

  1. the introduction lacks a clear explanation as to why graph-based methods are more suitable than DNNs for non-intrusive load monitoring. The comparison presented is only with conventional ML.

Response 2: Thanks a lot for the advice from the experts; we have revised it in detail.

From the experimental results in Table 2, we found that:

(1) On baseline datasets such as REDD and UK-Dale, the average result of the AttG-BDGNets method is better than other NILM methods and has a tremendous competitive advantage on most devices, but devices that handle fewer operations exist. Disadvantages, such as the MAE of the refrigerator in the REDD data is 3.52 higher than that of GCN, and the F1 value is 0.046 lower than that of GCN; it may be possible to further enhance the performance of rarely used devices by expanding the training sample size and improving the topology map construction strategy. On other devices, there may be two reasons for the better performance of our proposed method. First, the global features acquired by DynamicGCM and the context details extracted by AGM form a complementary relationship, enhancing the features' representation and making the method more efficient. Pay attention to the state of less-operated equipment; second, the designed weighted loss function, by updating network parameters to keep detailed information in time, improves the performance of less commonly used electrical equipment, such as kettles. In addition, the proposed AttG-BDGNets method takes devices as graph nodes and all attributes of each device as node features in constructing the topology graph. The edge weights between them highlight the direct differences between different devices and improve the final performance.

(2) The two most competitive methods are LDwA and GCN, and their average MAE on UK-Dale is 9.64 and 22.31,respectively. Among them, GCN also realizes prediction and classification by gathering node information; while LDwA is encoding - The decoding structure, CNN is used to obtain local features, and LSTM models the global semantics, thus prompting the network to improve the performance of less-operated devices. Similarly, our method is better than SGN on most devices, which means that our method is very beneficial for establishing complementary features for representation. At the same time, it also shows that this local attention guidance method can significantly refine features while Improved power consumption estimation and load classification, which improves overall performance. In more detail, compared with SGN, CNN, and BERT for the dataset REDD, the refrigerator MAE of our proposed AttG-BDGNets is reduced by 6.55, 17.01, and 11.24, respectively. On the f1 score, the classification performance of the refrigerator improved by 0.179, 0.178, and 0.076, respectively. In addition, our method outperformed DAE on all datasets, such as Dishwasher's F1 value improved by 0.155 and 0.153, respectively.

  1. it is unclear how the proposed method deals with temporal dependencies, particularly long-term ones. It is difficult to understand how GNNs identify and learn temporal dependencies or patterns.

Response 3: Thanks a lot for the advice from the experts; we have revised it in detail.

Graph convolution (GCN) [31] is a network that can directly perform convolution operations on topological graphs and mainly achieves feature capture by converging node features [27] The dynamic graph convolution module we designed not only learns node representation through node aggregation and transfer functions but also realizes dynamic aggregation between layer neighborhoods, which promotes the correlation between all devices and total power consumption, and in There is a correlation between the unmetered device and the metered device under test. In addition, a dependency relationship may also be formed between associated nodes that are far away.

Assume that the metered device under test is , the unmetered device is , the total power sequence of all devices is  ,is the sampling time, the sliding window length is , and the  power sequence is . Secondly, the sliding window is moved by  sampling time points. Finally, power sequences are obtained so that each power sequence can be represented as a topological graph, where there are   nodes in the graph, that is, devices contribute to each power sequence, nodes There is a correlation between them, and this correlation is called the edge weight between nodes.  To put it simply, we build a topology map with power sequences. Each power sequence has several electrical devices that generate power in different periods, and each device is a node of the topology graph. Formally, the topological graph  is shown in equation.

  1. the "Related works" section does not include any information on graph neural networks or their applications to non-intrusive load monitoring. It is recommended that the authors review GNN-based approaches in related fields.

Response 4: Thanks a lot for the advice from the experts; we have revised it in detail.

Faustine A et al. [28] considered that the performance of the device classifier is highly dependent on the signal characteristics of the load, but it is difficult to effectively distinguish between similar signals, which increases the difficulty of subsequent device classification, and proposed a weighted cycle graph (WRG) for Representing these signals improves the performance of device classification. Jiao X et al. [29] considered that many existing research methods have specific load decomposition errors when dealing with multi-modal devices. A non-intrusive load monitoring model based on a graph neural network is proposed. The graph structure is used to represent the relevant information between data nodes, combined with the long-term and short-term memory to extract the data's time-domain characteristics while retaining the power data's time characteristics. The correlation between different modes of equipment is improved, and the decomposition error is reduced. Jie Y et al. [30] proposed a non-intrusive load decomposition model based on a graph convolutional network (GCN) by fully mining the user's electricity consumption habits. The model first uses the time features extracted from the user's electricity consumption habits and constructs the power sequence as graph data based on the spectral graph theory as network input; then, based on the graph convolutional neural network, extracts the power attribute characteristics of each electrical appliance and its correlation Time-dependent structural features for non-intrusive load decomposition. These methods effectively improve the representation of key information in non-intrusive load monitoring tasks, but there is a lack of interaction between different information. At the same time, they aim to explore high-level semantic features while ignoring the rich semantics contained in low-level features so that when obtaining When the high-order features of the model cannot be effectively represented, it will affect the final recognition and decomposition accuracy.

  1. even in Chapter 3, there is no clear definition or explanation of how the method deals with temporal information and dependency. It is unclear how GCN interprets the data samples produced from different time intervals. Does the model focus only on graph structures for finding optimal feature representations? The data flow and learning strategy of the model, particularly regarding time, are too ambiguous.’’

Response 5: Thanks a lot for the advice from the experts; we have revised it in detail.

Graph convolution (GCN) [31] is a network that can directly perform convolution operations on topological graphs and mainly achieves feature capture by converging node features [27] The dynamic graph convolution module we designed not only learns node representation through node aggregation and transfer functions but also realizes dynamic aggregation between layer neighborhoods, which promotes the correlation between all devices and total power consumption, and in There is a correlation between the unmetered device and the metered device under test. In addition, a dependency relationship may also be formed between associated nodes that are far away.

Assume that the metered device under test is , the unmetered device is , the total power sequence of all devices is  ,is the sampling time, the sliding window length is , and the  power sequence is . Secondly, the sliding window is moved by  sampling time points. Finally, power sequences are obtained so that each power sequence can be represented as a topological graph, where there are   nodes in the graph, that is, devices contribute to each power sequence, nodes There is a correlation between them, and this correlation is called the edge weight between nodes.  To put it simply, we build a topology map with power sequences. Each power sequence has several electrical devices that generate power in different periods, and each device is a node of the topology graph. Formally, the topological graph  is shown in Eq.

AGM mainly consists of a bidirectional independent recurrent neural network (Bi-IndRNN) [25] and a local attention [26] guidance layer. Among them, Bi-IndRNN models the input NILM sequence from both positive and negative directions, aiming to explore the correlation between the total power and all devices and to construct the long-distance dependence between them in different periods. The traditional recurrent neural network (RNN) cannot effectively solve the problems of gradient disappearance and gradient explosion during the training process, and it isn't easy to learn long-term relationships. In addition, all neurons in the RNN layer are entangled, and their behavior is complex. Although the extended short-term memory network (LSTM) effectively solves these problems, using hyperbolic tangent and Sigmoid functions will cause the gradient to decay with the layer. Compared with RNN and LSTM neural networks, the neurons in the same layer of Bi-IndRNN are independent. The information between channels will spread and explore between multi-layer IndRNNs over time, and information can be cross-linked. Layer propagation improves the interpretability of neurons. At the same time, using unsaturated activation functions such as Relu solves the problem of gradient disappearance and explosion within and between layers, which enhances the robustness of the model and is excellent in processing long-term data more excellent.

Due to the differences in the power at different sampling time points and the state of the device, at the same time, the feature semantics (global and context details) at different levels have different emphases. Fusing these different feature semantics with a simple concatenation strategy brings only slight performance improvements and increases the use of redundant information. To fully use the feature information at each level, the global semantics obtained by the dynamic graph convolution module, and the contextual semantic details obtained by Bi-IndRNN, we designed a local attention guidance layer to enrich their semantics. This operation does not sacrifice. In the case of any details, it makes up for the lack of semantic representation of a single feature and forms an effective interaction. Furthermore, when one of the features of one party is underrepresented, to avoid excessive interference with this feature, we add a residual design to emphasize the importance of the features of the other party.

  1. the choice to adopt both MSE and KL in the loss function design requires further explanation. There is a concern that the proposed loss function may result in high residual for the learning model. The authors should explain the justification for the proposed loss function.

Response 6: Thanks a lot for the advice from the experts; we have revised it in detail.

To accurately predict the energy and simultaneously perform state classification, we design a weighted loss function that tunes and optimizes the network while minimizing absolute error. Among them,   can not only speed up the convergence speed but also reduce the gradient of the loss function as the error decreases, which helps the prediction result to be more stable. However, it is vulnerable to unmeasured discrete devices in the NILM sequence data or ignores the effects of these unmeasured devices, which leads to problems such as fitting or extraction explosion in the network during training; thus, the loss function is introduced to assist to adjust and optimize the model. The weighted loss function of  is shown in Eq.

In summary, we train the AttG-BDGNets method to generate predicted labels to match the actual labels and use a weighted loss function to optimize the proposed method. At the same time, the AdamW optimizer backpropagation is used to update the dynamic graph convolution module (DynamicGCM) Weight and bias parameters of the Attention Guidance Module (AGM); the following Algorithm 1 demonstrates the representation learning and training process of the proposed AttG-BDGNets method on NILM-related data.

Round 2

Reviewer 1 Report

Accept.

Author Response

Point 1: Accept.

Response 1: Many thanks to the experts for their suggestions, which we have revised in detail.

Reviewer 2 Report

The authors have responded to all inquiries and comments regarding the manuscript. The topic under discussion is intriguing, and a number of noteworthy studies are presented. Thus, it is our opinion that this manuscript merits publication in its present form.

Author Response

Point 1: The authors have responded to all inquiries and comments regarding the manuscript. The topic under discussion is intriguing, and a number of noteworthy studies are presented. Thus, it is our opinion that this manuscript merits publication in its present form.

Response 1: Many thanks to the experts for their suggestions, which we have revised in detail.

Reviewer 3 Report

The paper has been well revised to reflect the detailed opinions. However, it appears to be better to plot Loss and Acc separately in Figure 2, rather than together. This is because Acc has a range of 0 to 1, while it is difficult to know the maximum value of Loss. Of course, Loss can also be represented as Acc by using Min-max scaling, but it is recommended to plot them separately because it is difficult to capture sensitive changes when the Loss value becomes small.

Author Response

The Response have already provided in the PDF file below.
